# Pressure Control of Multi-Mode Variable Structure Electro–Hydraulic Load Simulation System

**DOI:** 10.3390/s24227400

**Published:** 2024-11-20

**Authors:** He Hao, Hao Yan, Qi Zhang, Haoyu Li

**Affiliations:** 1School of Mechanical, Electronic and Control Engineering, Beijing Jiaotong University, Beijing 100044, China; 22121296@bjtu.edu.cn (H.H.); lhy10102021@163.com (H.L.); 2AECC Guizhou Honglin Aero–Engine Control Technology Co., Ltd., Guiyang 551522, China; qizhang821@163.com

**Keywords:** independent load port, electro–hydraulic servo system, sliding mode control, pressure control

## Abstract

During the loading process, significant external position disturbances occur in the electro–hydraulic load simulation system. To address these position disturbances and effectively mitigate the impact of uncertainty on system performance, this paper first treats model parameter uncertainty and external disturbances as lumped disturbances. The various states of the servo valve and the pressures within the hydraulic cylinder chambers are then examined. Building on this foundation, the paper proposes a nonlinear multi-mode variable structure independent load port electro–hydraulic load simulation system that is tailored for specific loading conditions. Secondly, in light of the significant motion disturbances present, this paper proposes an integral sliding mode active disturbance rejection composite control strategy that is based on fixed-time convergence. Based on the structure of the active disturbance rejection control framework, the fixed-time integral sliding mode and active disturbance rejection control algorithms are integrated. An extended state observer is designed to accurately estimate the lumped disturbance, effectively compensating for it to achieve precise loading of the independent load port electro–hydraulic load simulation system. The stability of the designed controller is also demonstrated. The results of the simulation research indicate that when the command input is a step signal, the pressure control accuracy under the composite control strategy is 99.94%, 99.86%, and 99.76% for disturbance frequencies of 1 Hz, 3 Hz, and 5 Hz, respectively. Conversely, when the command input is a sinusoidal signal, the pressure control accuracy remains high, measuring 99.94%, 99.8%, and 99.6% under the same disturbance frequencies. Furthermore, the simulation demonstrates that the influence of sensor random noise on the system remains within acceptable limits, highlighting the effective filtering capabilities of the extended state observer. This research establishes a solid foundation for the collaborative control of load ports and the engineering application of the independent load port electro–hydraulic load simulation system.

## 1. Introduction

Electro–hydraulic servo systems have the advantages of fast responses, high control accuracy, and easy equipment maintenance and have been widely used in industry, aerospace, and other fields [1,2]. The design of the motion controller for the servo system is challenging due to numerous model uncertainties [3,4], including parameter uncertainties, unmodeled dynamics, and external disturbances. This complexity represents a significant topic of current research.

To enhance the control performance of electro–hydraulic servo systems, numerous advanced control strategies have been proposed, and extensive research has been conducted by scholars from various countries. Generally speaking, robust control [5,6] can ensure the specified output tracking performance but requires a large feedback gain to ensure high control accuracy in practical applications. Adaptive control [7,8] can aptly solve the problem of system parameter uncertainty and achieve the online estimation of unknown parameters by designing parameter adaptive laws. The adaptive robust control strategy [9,10] develops nonlinear robust terms to effectively counter external disturbances while simultaneously reducing feedback gain through the adaptive adjustment of system parameters. This approach ensures optimal transient and steady-state control performance. Adaptive disturbance rejection control [11,12] integrates the benefits of both adaptive control and active disturbance rejection control. This approach reduces the adaptive feedback gain by designing the total disturbance of the uncertainty within the extended state observer (ESO) model and compensating for it. Due to its excellent approximation characteristics, the neural network [13,14] possesses adaptive and self-learning capabilities, enabling it to perform nonlinear compensation and model identification. As a result, it is increasingly employed to address nonlinear problems within various systems. Sliding mode control [15,16] can theoretically achieve asymptotic tracking control but usually results in discontinuous control inputs and jitter problems in the physical system. Model predictive control [17,18] has been favored by many scholars due to its ability to explicitly handle constraints and system uncertainties and has been widely used in electromechanical servo systems. The tracking error of the servo system must not only achieve a specified steady-state accuracy but must also adhere to certain transient performance indicators, such as convergence speed and overshoot. Preset performance control [19,20] offers an improved solution for enhancing the transient performance of the system.

The traditional hydraulic system employs a single-spool servo valve to regulate the oil port of the hydraulic actuator. Although this configuration meets the desired control performance, it results in repeated throttling losses, which complicates the optimization of energy-saving characteristics. To address the recurring throttling losses prevalent in traditional hydraulic systems, BACKÉ [21] proposed an independent load port control system. This system employs four cartridge valves to independently regulate the inlet and outlet oil circuits of the hydraulic actuator, thereby facilitating both the positive and negative aspects of motion control for the hydraulic actuator. On this basis, many scholars from various countries have conducted research on principle configuration, motion control, fault diagnosis, and other aspects [22,23,24,25,26]. To reduce pressure losses, Li et al. [27] proposed an independent metering valve-based configuration driven by servo pump control. Wang et al. [28], pursuing efficiency improvement, proposed a differential drive collaborative (DDCS) control for vehicles with independent-wheel drive. In another paper, Kogler et al. [29] realized an IM concept of open-loop control cylinder drive, which was combined with a human operator, constituting the so-called cybernetic system (CPS). Nguyen et al. [30] implemented a novel independent metering valve system (NIMV) to reduce energy consumption. Koivumäki et al. [31] used a separate meter-in–meter-out (SMISMO) setup, enabling the independent metering (pressure control) of each chamber actuator. Abuowda et al. [32] reviewed independent metering advanced technology used in hydraulic systems.

Based on the independent load port as a novel valve group control technology, this valve group employs dual valve cores to independently regulate the throttling area of both the inlet and outlet. This design eliminates the rigid linkage between the two chambers of the hydraulic cylinder, thereby enhancing the degree of control freedom. Furthermore, it allows for the separate control of pressure in the two chambers of the hydraulic cylinder, facilitating output loading while simultaneously considering energy-saving performance. In view of this, this paper proposes an independent load port electro–hydraulic load simulation system. To address the challenges posed by uncertain nonlinearity, unmeasurable states, and loading accuracy in the electro–hydraulic load simulation system, a complex multi-mode variable structure mathematical model was initially developed for the independent load port electro–hydraulic load simulation system, which is subject to disturbances from external motion. The traditional Active Disturbance Rejection Control (ADRC) framework exhibits certain limitations, including inadequate tracking performance in the loading control of electro–hydraulic load simulation systems and the occurrence of chattering in sliding mode control. To address these issues, this paper introduces an integral sliding mode active disturbance rejection composite control strategy based on fixed-time convergence, referred to as FISMADRC. By designing an extended state observer, the system state and external disturbances can be accurately estimated, thereby addressing the unmeasured aspects of the system state while simultaneously compensating for the effects of time-varying disturbances on control performance. The effectiveness of the proposed independent load port electro–hydraulic load simulation system, along with the designed composite controller, is demonstrated through simulation comparisons.

## 2. Variable Structure Independent Load Port Model

The working principle of the independent load port electro–hydraulic load simulation system is illustrated in Figure 1. On the left side, the electro–hydraulic loading system comprises a loading servo valve, a loading cylinder, and a pressure sensor, among other components, and operates within a closed-loop feedback mechanism. Conversely, the right side features the linear actuator position system, which also employs a closed-loop feedback system facilitated by a displacement sensor. The electro–hydraulic loading system is rigidly connected to the linear actuator. Within this system, the displacement of the linear actuator represents a significant position disturbance to the loading system. The independent load port valve control system utilizes control technology that allows for the independent adjustment of the throttling areas of both the inlet and outlet valves. To investigate the fundamental theory of the independent load port electro–hydraulic load simulation system, this article intentionally excludes considerations regarding the pressure coordination of the hydraulic cylinder’s right chamber and the control of the output force of the hydraulic cylinder. In this context, $u_{1}$ represents the input signal of the servo valve, $p_{s}$ denotes the system pressure, $p_{1}$ refers to the pressure in the left chamber of the hydraulic cylinder, and $p_{1}$ is also the control target examined in this study.

In this article, the right chamber of the hydraulic cylinder is connected to the oil-return pipeline, while the servo valve manages the pressure control of the electro–hydraulic load simulation system via an independent load port. The pressure variations in the left chamber of the hydraulic cylinder result in complex working conditions for the independent load port electro–hydraulic load simulation system, characterized by multi-mode variable structures. In working condition 1, the system is considered to be under normal operating conditions when the system pressure $p_{S}$ exceeds the pressure $p_{1}$ in the left chamber of the hydraulic cylinder. In working condition 2, when the spool of the servo valve is actuated in the forward direction but the system pressure $p_{S}$ falls below the pressure $p_{1}$ in the left chamber, the oil within the left chamber is forcibly discharged through the oil inlet of the servo valve, resulting in a release of pressure. In working condition 3, when the piston rod of the hydraulic cylinder is subjected to an external force, the pressure $p_{1}$ in the left chamber experiences a rapid increase, significantly surpassing the system pressure $p_{S}$. Consequently, the spool of the servo valve opens in the reverse direction, allowing the oil from the left chamber to flow back to the tank.

According to the working principle of the multi-mode variable structure of the independent load port electro–hydraulic load simulation system, a nonlinear mathematical model of the system is established. Equation (1) represents the flow equation of the servo valve when the spool is opened in the forward direction and the system pressure $p_{S}$ exceeds the chamber pressure $p_{1}$. Equation (2) describes the flow equation of the servo valve when the spool is again opened in the forward direction, but in this case, the chamber pressure $p_{1}$ is greater than the oil source pressure $p_{S}$. Finally, Equation (3) outlines the flow equation of the servo valve when the valve core is opened in the reverse direction, specifically when the chamber pressure $p_{1}$ surpasses the return oil back pressure.
(1)$$\left\{\begin{array}{c} x_{v}>0,p_{s}>p_{1}\\ Q_{1}=C_{d}wx_{v}\sqrt{\frac{2\left(p_{s}-p_{1}\right)}{\rho }} \end{array}\right.$$
(2)$$\left\{\begin{array}{c} x_{v}>0,p_{s}<p_{1}\\ Q_{1}=-C_{d}wx_{v}\sqrt{\frac{2\left(p_{1}-p_{s}\right)}{\rho }} \end{array}\right.$$
(3)$$\left\{\begin{array}{c} x_{v}<0\\ Q_{1}=C_{d}wx_{v}\sqrt{\frac{2p_{1}}{\rho }} \end{array}\right.$$
where $Q_{1}$ represents the volume flow rate entering the chamber, $C_{d}$ denotes the valve port flow coefficient, w indicates the valve opening area gradient, $p_{S}$ refers to the system pressure, ρ signifies the oil density, and $p_{1}$ is the pressure in the left chamber.

For a symmetrical hydraulic cylinder, assuming that the initial volumes of the two chambers loading the cylinder are equal and neglecting external leakage from the cylinder, the flow continuity equation for the pressure chamber is as follows.
(4)$$Q_{1}=A\frac{dy}{dt}+\frac{V_{1}}{\beta_{e}}\frac{dp_{1}}{dt}+C_{tc}p_{1}$$
where y represents the displacement of the hydraulic cylinder piston rod, A denotes the effective area of the piston, $V_{1}$ indicates the total control volume, which encompasses both the drive chamber and the servo valve pipeline, $\beta_{e}$ refers to the liquid volume elastic modulus of the hydraulic fluid, and $C_{tc}$ signifies the internal leakage coefficient of the hydraulic cylinder.

We assume that the system state variable is $x_{1}=p_{1}$ and the control variable $u=x_{v}$. According to Equations (1)–(4), the state equations under different working conditions can be obtained.
(5)$$\left\{\begin{array}{c} u>0,p_{s}>p_{1}\\ {\dot{x}}_{1}=\frac{-\beta_{e}C_{tc}}{V_{1}+Ay}x_{1}+\frac{\beta_{e}C_{d}w}{V_{1}+Ay}\sqrt{\frac{2\left(p_{s}-x_{1}\right)}{\rho }}u-\frac{\beta_{e}A}{V_{1}+Ay}\dot{y} \end{array}\right.$$
(6)$$\left\{\begin{array}{c} u>0,p_{s}<p_{1}\\ {\dot{x}}_{1}=\frac{-\beta_{e}C_{tc}}{V_{1}+Ay}x_{1}-\frac{\beta_{e}C_{d}w}{V_{1}+Ay}\sqrt{\frac{2\left(x_{1}-p_{s}\right)}{\rho }}u-\frac{\beta_{e}A}{V_{1}+Ay}\dot{y} \end{array}\right.$$
(7)$$\left\{\begin{array}{c} u<0\\ {\dot{x}}_{1}=\frac{-\beta_{e}C_{tc}}{V_{1}+Ay}x_{1}+\frac{\beta_{e}C_{d}w}{V_{1}+Ay}\sqrt{\frac{2x_{1}}{\rho }}u-\frac{\beta_{e}A}{V_{1}+Ay}\dot{y} \end{array}\right.$$

## 3. Controller Design

This study presents the design of a controller for the left chamber pressure $p_{1}$ of the independent load port electro–hydraulic load simulation system. After analyzing and organizing Equations (5)–(7), the state equation for the independent load port electro–hydraulic load simulation system can be expressed in a unified form.
(8)$$\left\{\begin{array}{c} {\dot{x}}_{1}=f(x_{1})+bu\\ y_{1}=x_{1} \end{array}\right.$$
where $y_{1}$ is the output of the system state variable, $x_{1}$ is the state variable of the system, $f(x_{1})$ is an item that has nothing to do with the control quantity, u is the control input of the system, and b is the control compensation factor.

It can be seen from state Equations (5)–(7) of the valve-controlled independent load port electro–hydraulic load simulation system that $f(x_{1})$ is as follows:(9)$$f(x_{1})=-\frac{\beta_{e}A}{V_{1}+Ay}\dot{y}-\frac{\beta_{e}C_{ec}}{V_{1}+Ay}x_{1}$$

It can be seen that it contains motion disturbances and some parameters that may be uncertain. Under different working conditions, b can be written in the following form:(10)$$\left\{\begin{array}{l} b=\frac{\beta_{e}C_{d}w}{V_{1}+Ay}\sqrt{\frac{2\left(p_{s}-x_{1}\right)}{\rho }},u>0,p_{s}>p_{1}\\ b=\frac{-\beta_{e}C_{d}w}{V_{1}+Ay}\sqrt{\frac{2\left(x_{1}-p_{s}\right)}{\rho },}u>0,p_{s}<p_{1}\\ b=\frac{\beta_{e}C_{d}w}{V_{1}+Ay}\sqrt{\frac{2x_{1}}{\rho }},u<0 \end{array}\right.$$

In this context, the control item part, denoted as bu, is influenced by the variable b, which exhibits certain fluctuations. Consequently, the changing component of b can be considered a disturbance, while the constant component can be represented as $b_{0}$. It is important to note that the value of $b_{0}$ is not unique.
(11)$$bu=\Delta bu+b_{0}u$$
where Δb is the uncertainty of the input gain.

We define the tracking error of the left chamber pressure $p_{1}$ of the independent load port electro–hydraulic load simulation system as follows:(12)$$e(t)=p_{r1}(t)-p_{1}(t)$$

Among them, $p_{r1}(t)$ is the expected pressure of the left chamber.

For the independent load port electro–hydraulic load simulation system, the pressure-fixed time integral sliding mode surface of the left cavity is defined as follows:(13)$$s(t)=e(t)+c_{1}\int_{0}^{t}{\left|e\right|}^{m}\;sign(e)dt+c_{2}\int_{0}^{t}{\left|e\right|}^{n}\;sign(e)dt$$
where $c_{1}>0$, $c_{2}>0$ is the design gain, m>1, 0<n<1.

The first-order differential of the fixed-time integral sliding mode surface of the pressure in the left cavity is as follows:(14)$$\dot{s}(t)={\dot{p}}_{r1}(t)-f(x_{1})-bu+c_{1}{\left|e\right|}^{m}\;sign(e)+c_{2}{\left|e\right|}^{n}\;sign(e)$$

We select the fixed-time exponential approaching law as follows:(15)$$\dot{s}(t)=-\varepsilon sign(s)-g_{1}{\left|s\right|}^{r}\;sign(s)-g_{2}{\left|s\right|}^{\varphi }\;sign(s)$$
where ε>0, $g_{1}>0$, $g_{2}>0$, r>0, φ>0.

Connecting Equations (14) and (15), the pressure control rate of the left chamber of the hydraulic cylinder can be obtained.
(16)$$u=\frac{1}{b}\left[\begin{array}{l} {\dot{p}}_{r1}(t)-f(x_{1})+c_{1}{\left|e\right|}^{m}\;sign(e)+c_{2}{\left|e\right|}^{n}\;sign(e)+\varepsilon sign(s)\\ +g_{1}{\left|s\right|}^{r}\;sign(s)+g_{2}{\left|s\right|}^{\varphi }\;sign(s) \end{array}\right]$$
where $f(x_{1})$ is estimated by the expanded state observer observation of Equation (18).

The expanded state system equation of the system’s state Equation (8) can be written as follows:(17)$$\left\{\begin{array}{c} \begin{array}{l} {\dot{x}}_{1}=x_{2}+b_{0}u\\ {\dot{x}}_{2}=d\left(t\right) \end{array}\\ y_{1}=x_{1} \end{array}\right.$$

An expanded state observer of the following form is used:(18)$$\left\{\begin{array}{c} e=z_{1}-y_{1}\\ {\dot{z}}_{1}=z_{2}-\beta_{1}e+b_{0}u\\ {\dot{z}}_{2}=-\beta_{2}\mathrm{fal}(e,\alpha ,\delta ) \end{array}\right.$$
where $z_{1}$ is the observed value of $x_{1}$, $z_{2}$ is the observed value of $x_{2}$, $\beta_{1}$ and $\beta_{2}$ are appropriate parameters, e is the pressure error value, δ is the interval length of the linear segment, α is a real number from 0 to 1, and fal is a continuous power function with linear segments, as shown in (19), and is used to avoid high-frequency chatter.
(19)$$\mathrm{fal}(e,\alpha ,\delta )=\left\{\begin{array}{c} e/\delta^{a-1},\left|e\right|\leq \delta \\ {\left|e\right|}^{a}\;sign(e),\left|e\right|>\delta \end{array}\right.$$

Due to the observation error inherent in the extended state observer, stability issues arise in the inner loop of active disturbance rejection control, which in turn influences the permissible range of $b_{0}$. If the value of $b_{0}$ is excessively small, the inner control loop of active disturbance rejection may diverge as a result of an overly high equivalent open-loop gain. Conversely, if $b_{0}$ is too large, significant control errors may occur. Considering these factors comprehensively, this paper adopts the invariant component $b_{0}$ of b, as specified in Equation (20), and substitutes it with a constant during the simulation process.
(20)$$b_{0}=\beta_{e}C_{d}\omega \sqrt{2p_{s}}/2V_{1}\sqrt{\rho }=35,224,325\;(\mathrm{kPa}\cdot s)/m$$

The structural block diagram of the fixed-time convergence integral sliding mode active disturbance rejection composite controller designed in this section is shown in Figure 2.

Before performing an analysis of the fixed-time convergent integral sliding mode active disturbance rejection composite controller, the following lemma is first given:

**Lemma** **1.** *For a class of nonlinear systems*.


(21)
$$\dot{x}(t)=f(t)$$


Among them, it is assumed that there is a Lyapunov function V(x) that satisfies.
(22)$$\dot{V}(x)\leq -\gamma_{1}V^{\delta_{1}}(x)-\gamma_{2}V^{\delta_{2}}(x)$$

Among these parameters, if the values satisfy conditions $\gamma_{1}>0$, $\gamma_{2}>0$, $0<\delta_{1}<1$, $\delta_{2}>1$, the system is classified as globally fixed-time stable. In this case, the convergence time *T* is bounded, and it adheres to the inequality presented in Equation (23).
(23)$$T\leq T_{\max}=\frac{1}{\gamma_{1}(1-\delta_{1})}+\frac{1}{\gamma_{2}(\delta_{2}-1)}$$

In order to prove that the fixed-time integrated sliding mode surface s(t) can converge to zero in a fixed time, the Lyapunov function is selected as follows:(24)$$V_{1}=\frac{1}{2}s^{2}$$

Derive Equation (24) and organize it to obtain
(25)$${\dot{V}}_{1}=s\dot{s}=s[{\dot{p}}_{r1}(t)-f(x_{1})-bu+c_{1}{\left|e\right|}^{m}\;sign(e)+c_{2}{\left|e\right|}^{n}\;sign(e)]$$

Substituting the designed control rate (16) into Equation (25), the above equation can be continued to be organized as follows:(26)$$\begin{array}{l} {\dot{V}}_{1}=s[-\varepsilon sign(s)-g_{1}{\left|s\right|}^{r}\;sign(s)-g_{2}{\left|s\right|}^{\varphi }\;sign(s)]\\ \leq s[-g_{1}{\left|s\right|}^{r}\;sign(s)-g_{2}{\left|s\right|}^{\varphi }\;sign(s)]\\ =-g_{1}{\left(s^{2}\right)}^{\frac{1}{2}(r+1)}-g_{2}{\left(s^{2}\right)}^{\frac{1}{2}(\varphi +1)}\\ =-g_{1}{\frac{1}{2}}^{\frac{1}{2}(r+1)}{\left(V_{1}\right)}^{\frac{1}{2}(r+1)}-g_{2}{\frac{1}{2}}^{\frac{1}{2}(\varphi +1)}{\left(V_{1}\right)}^{\frac{1}{2}(\varphi +1)}<0 \end{array}$$

It is evident that Equation (26) is always less than zero. According to Lemma 1, the designed fixed-time convergent integral sliding mode surface s(t) meets the reachability condition, indicating that the fixed-time (27) is asymptotically stable.
(27)$$T_{s1}\leq \frac{1}{g_{1}{\frac{1}{2}}^{\frac{1}{2}(r+1)}(1-\frac{1}{2}(r+1))}+\frac{1}{g_{2}{\frac{1}{2}}^{\frac{1}{2}(\varphi +1)}(\frac{1}{2}(\varphi +1)-1)}$$

To prove the convergence of the tracking error of the system state, we refer to Equation (27), which indicates that the integral sliding mode variable converges in a fixed timeframe. When the integral sliding mode variable reaches stability, denoted as $s=\dot{s}=0$, we can derive the following results:(28)$$\dot{e}=-c_{1}{\left|e\right|}^{m}\;sign(e)-c_{2}{\left|e\right|}^{n}\;sign(e)$$

We then select the following Lyapunov function:(29)$$V_{2}=\frac{1}{2}e^{2}$$

By taking the derivative of Equation (29) and sorting it out, we can obtain
(30)$$\begin{array}{l} V_{2}=e\dot{e}=e\left[-c_{1}{\left|e\right|}^{m}\;sign(e)-c_{2}{\left|e\right|}^{n}\;sign(e)\right]\\ =-c_{1}{\left(e^{2}\right)}^{\frac{1}{2}(m+1)}-c_{2}{\left(e^{2}\right)}^{\frac{1}{2}(n+1)}\\ =-c_{1}{\frac{1}{2}}^{\frac{1}{2}(m+1)}{\left(V_{2}\right)}^{\frac{1}{2}(m+1)}-c_{2}{\frac{1}{2}}^{\frac{1}{2}(n+1)}{\left(V_{2}\right)}^{\frac{1}{2}(n+1)}<0 \end{array}$$

It can be observed that Equation (30) is consistently less than zero. From Lemma 1, it is evident that the tracking error of the system, governed by the integrated sliding mode active disturbance rejection composite control strategy with fixed-time convergence, converges in a fixed time. Specifically, the fixed time (31) is asymptotically stable.
(31)$$T_{s2}\leq \frac{1}{c_{1}{\frac{1}{2}}^{\frac{1}{2}(m+1)}(1-\frac{1}{2}(m+1))}+\frac{1}{c_{2}{\frac{1}{2}}^{\frac{1}{2}(n+1)}(\frac{1}{2}(n+1)-1)}$$

## 4. Simulation Results and Analysis

### 4.1. Closed-Loop Pressure Simulation Analysis

It is assumed that the external position disturbance experienced by the independent load port electro–hydraulic load simulation system is characterized by sinusoidal motion. During the simulation process, the position disturbance signal is designated as $x_{d}=0.01\sin(2\pi t)$ (m), with an amplitude of 10 mm. Since the frequency of external position disturbances is typically low, a frequency range of 1–5 Hz is selected as it is commonly utilized in engineering practice. If the frequency value is excessively high, the system will demand significant energy, necessitating corresponding enhancements to the servo valve and energy system. Additionally, the displacement of the hydraulic cylinder will be minimal, resulting in observable vibrations without noticeable displacement.

Under the influence of sinusoidal motion disturbances in the independent load port of the electro–hydraulic load simulation system, a comparative simulation study was conducted on fixed-time integral sliding mode active disturbance rejection composite control, PID control, and active disturbance rejection control strategies. The system command signals are designated as $x_{1}=5000\;\mathrm{kPa}$ and $x_{1}=5000\sin(2\pi t)+8000\;\mathrm{kPa}$, respectively. The parameter selection and simulation parameters for the system are detailed in Table 1 and Table 2. The closed-loop output pressure and the output pressure error of the system are illustrated in the figure below.

According to the closed-loop pressure curve in Figure 3, the closed-loop pressure error curve in Figure 4, and the controller output in Figure 5, we illustrate that when the disturbance signal frequency is set to 1 Hz, the composite control of fixed-time integral sliding mode auto-disturbance rejection results in a rise time of approximately 0.03 s for the pressure to reach its steady-state value. Notably, prior to achieving a steady state, the pressure exhibits significant fluctuations at 0.01 s. Following this initial period, the pressure stabilizes without overshooting, transitioning smoothly to the steady-state pressure. Once the pressure attains the steady-state condition, it closely aligns with the command pressure curve. The steady-state pressure shows minimal adjustment, with a steady-state error of approximately 3 kPa, indicating a pressure control accuracy of 99.94%. Under active disturbance rejection control, the rise time for the pressure to reach the steady-state value is approximately 0.035 s. There is a fluctuation observed at 0.0025 s as the pressure approaches the steady-state value, followed by a smooth transition to steady-state pressure. Once steady-state pressure is attained, the pressure overshoots by approximately 12 kPa, while still maintaining a high level of accuracy with a pressure control accuracy of 99.76%. Under PID control, the rise time for the pressure to reach a steady state is approximately 0.06 s. The pressure exhibits continuous fluctuations prior to achieving a steady state. Once a steady state is attained, the maximum pressure overshoot is recorded at 140 kPa and the pressure control accuracy is measured at 97.2%. In comparison, the fixed-time integral sliding mode active disturbance rejection composite control demonstrates a pressure response that is 0.005 s and 0.03 s faster than the active disturbance rejection control and PID pressure control, respectively.

According to the closed-loop pressure curve in Figure 6, the closed-loop pressure error curve in Figure 7, and the controller output in Figure 8, we illustrate that when the disturbance signal frequency is set to 3 Hz, the composite control of fixed-time integral sliding mode auto-disturbance rejection results in a rise time of approximately 0.03 s for the pressure to attain the steady-state value. Prior to reaching this steady-state pressure, fluctuations occur within a duration of 0.01 s, followed by a smooth transition to the steady state without any overshoot. Once the pressure stabilizes, the overshoot is minimal and the steady-state pressure error is approximately 7 kPa, indicating a high-pressure control accuracy of 99.86%. Under active disturbance rejection control, the rise time for the pressure to reach the steady-state value is approximately 0.035 s. There is an initial fluctuation before the pressure stabilizes, followed by a transition to the steady-state pressure without any overshoot. Once at a steady state, the pressure exhibits an overshoot of approximately 15 kPa while maintaining a high level of accuracy, with a pressure control accuracy of 99.7%. Under PID control, the rise time for the pressure to reach a steady state is approximately 0.06 s; however, the pressure consistently fluctuates. Once steady-state pressure is achieved, the maximum pressure overshoot reaches 150 kPa and the pressure control accuracy is 97%. In comparison, the fixed-time integral sliding mode active disturbance rejection composite control demonstrates a pressure response that is 0.005 s and 0.03 s faster than the active disturbance rejection control and PID pressure control, respectively.

According to the closed-loop pressure curve in Figure 9, the closed-loop pressure error curve in Figure 10, and the controller output in Figure 11, we illustrate that when the disturbance signal frequency is set at 5 Hz, the composite control employing fixed-time integral sliding mode auto-disturbance rejection results in a rise time of approximately 0.03 s for the pressure to reach the steady-state value. Notably, the pressure experiences some fluctuations of approximately 0.01 s before stabilizing, subsequently transitioning smoothly to the steady-state pressure without any overshoot. Once the steady-state pressure is attained, the overshoot remains minimal, with a steady-state error of about 12 kPa, resulting in a pressure control accuracy of 99.76%. Under active disturbance rejection control, the rise time for the pressure to reach the steady-state value is approximately 0.035 s. Notably, there is a fluctuation prior to the pressure stabilizing at the steady-state level, after which it transitions smoothly. Once steady-state pressure is achieved, the pressure experiences an overshoot of approximately 25 kPa while maintaining a high level of accuracy with a pressure control precision of 99.5%. Under PID control, the rise time for the pressure to reach the steady-state value is approximately 0.06 s. There is consistently some vibration upon reaching steady-state pressure. Once steady-state pressure is achieved, the maximum pressure overshoot recorded is 160 kPa and the pressure control accuracy is 96.8%. In comparison, the fixed-time integral sliding mode active disturbance rejection composite control demonstrates a pressure response that is 0.005 s and 0.03 s faster than the active disturbance rejection control and PID pressure control, respectively.

According to the closed-loop pressure curve in Figure 12, the closed-loop pressure error curve in Figure 13, and the controller output in Figure 14, when the frequency of the disturbance signal is set to 1 Hz, the closed-loop pressure curve exhibits a rapid response under the fixed-time integral sliding mode active disturbance rejection composite control. The dynamic tracking performance of the closed-loop pressure is commendable, with a tracking time of approximately 0.035 s. The closed-loop pressure curve effectively follows the command pressure, accurately tracking the peaks and troughs of the sinusoidal pressure curve. In this state, the transition of the sinusoidal pressure curve is smooth and stable, exhibiting no buffeting. As illustrated in Figure 13, the maximum overshoot of the pressure upon reaching the steady state is approximately 3 kPa, resulting in a pressure control accuracy of 99.94%. Under active disturbance rejection control, the pressure exhibits a rapid response; however, there is a noticeable lag in pressure tracking, with a tracking time of approximately 0.045 s, leading to a specific tracking error in the pressure curve. As illustrated in Figure 13, the maximum overshoot of the pressure upon reaching the steady state is approximately 138 kPa, and the pressure control accuracy is 97.24%. Under PID control, the closed-loop pressure curve exhibits a slightly sluggish response, resulting in a delay in tracking the pressure curve. The closed-loop pressure aligns with the command pressure at approximately 0.08 s. Due to the absence of filtering in PID control, noise significantly affects the pressure curve, particularly at its peaks and troughs. As illustrated in Figure 13, the maximum overshoot of the pressure upon reaching a steady state is approximately 154 kPa, with a pressure control accuracy of 96.92%. In comparison, the fixed-time integral sliding mode active disturbance rejection composite control demonstrates a pressure response that is 0.01 s and 0.045 s faster than the active disturbance rejection control and PID pressure control, respectively.

According to the closed-loop pressure curve in Figure 15, the closed-loop pressure error curve in Figure 16, and the controller output in Figure 17, when the frequency of the disturbance signal is set to 3 Hz, the closed-loop pressure curve exhibits a rapid response under the fixed-time integral sliding mode active disturbance rejection composite control, tracking the command pressure in approximately 0.035 s. The dynamic tracking performance of the closed-loop pressure is commendable and the pressure curve transitions smoothly and stably at the peaks and troughs of the sinusoidal pressure curve, without exhibiting any buffeting. Furthermore, as illustrated in Figure 16, the maximum overshoot upon reaching steady-state pressure is approximately 10 kPa, resulting in a pressure control accuracy of 99.8%. Under active disturbance rejection control, the pressure exhibits a rapid response; however, a hysteresis phenomenon is observed, with a tracking time of approximately 0.045 s, leading to a certain tracking error in the pressure curve. As illustrated in Figure 16, the maximum overshoot of the pressure upon reaching the steady state is approximately 140 kPa and the pressure control accuracy is 97.2%. Under PID control, the closed-loop pressure curve exhibits a slight delay, with the pressure curve aligning with the upper command pressure at 0.08 s. Due to the lack of filtering in PID control, noise significantly affects the pressure curve, particularly at its peaks and troughs. As illustrated in Figure 16, the maximum overshoot of the pressure upon reaching a steady state is approximately 160 kPa, resulting in a pressure control accuracy of 96.8%. In comparison, the fixed-time integral sliding mode active disturbance rejection composite control demonstrates a pressure response that is 0.01 s and 0.045 s faster than the active disturbance rejection control and PID pressure control, respectively.

According to the closed-loop pressure curve in Figure 18, the closed-loop pressure error curve in Figure 19, and the controller output in Figure 20, when the frequency of the disturbance signal is set to 5 Hz, the closed-loop pressure curve exhibits a rapid response and successfully tracks the upper command pressure within approximately 0.035 s, demonstrating effective dynamic tracking performance. The pressure curve accurately follows the peaks and troughs of the sinusoidal pressure profile, transitioning smoothly and stably without any noticeable oscillations. Furthermore, as indicated in Figure 19, the maximum overshoot of the pressure upon reaching the steady state is approximately 20 kPa, while the pressure control accuracy is measured at 99.6%. Under active disturbance rejection control, a hysteresis phenomenon occurs in pressure tracking, with a tracking time of approximately 0.045 s, leading to a measurable tracking error in pressure. As illustrated in Figure 19, the maximum overshoot of pressure upon reaching a steady state is approximately 150 kPa and the pressure control accuracy is 97%. Under PID control, the closed-loop pressure curve exhibits a slight delay, with the pressure curve tracking the upper command pressure at 0.08 s. Due to the lack of filtering in the PID controller, noise significantly affects the pressure curve, particularly at its peaks and troughs. As illustrated in Figure 19, the maximum overshoot of the pressure upon reaching the steady state is approximately 170 kPa, resulting in a pressure control accuracy of 96.6%. In comparison, the fixed-time integral sliding mode active disturbance rejection composite control demonstrates a pressure response that is 0.01 s and 0.045 s faster than the active disturbance rejection control and PID pressure control, respectively.

In the loading control of the electro–hydraulic load simulation system, an accuracy of 1% in pressure control error is considered excellent. This study employs a fixed-time convergence integral sliding mode active disturbance rejection composite control strategy. When the command is a step signal of $x_{1}=5000\mathrm{kPa}$ and the disturbance frequencies are set to 1 Hz, 3 Hz, and 5 Hz, the pressure control error accuracy achieved under the composite control strategy is 0.06%, 0.14%, and 0.24%, respectively. When the command is a sinusoidal signal of $x_{1}=5000\sin(2\pi t)+8000\mathrm{kPa}$, the pressure control accuracy under the same disturbance frequency is measured at 0.06%, 0.2%, and 0.4%, respectively. It is evident that the pressure control error of the multi-mode variable structure electro–hydraulic load simulation system remains consistently controlled within 1%. The simulation analysis of closed-loop pressure indicates that, under the influence of identical command and disturbance signals, the fixed-time integral sliding mode active disturbance rejection composite control achieves a faster rise time in reaching steady-state pressure compared to both active disturbance rejection and PID control. Furthermore, the overshoot of the state error is minimal and the pressure control accuracy is higher. These results demonstrate that the pressure control performance and robustness of the fixed-time integral sliding mode active disturbance rejection composite control are superior.

### 4.2. Noise Impact Analysis

The interference of noise significantly affects the control performance of system pressure. This paper investigates the filtering capabilities of the extended state Observer (ESO) in scenarios where the pressure sensor is subject to noise, as encountered in the actual operating conditions of the independent load port electro–hydraulic load simulation system. During the simulation, the pressure sensor is converted into a practical 0–10 V analog pressure sensor, with the specified system command signal set at $x_{1}=5000\sin(2\pi t)+6000$ and the disturbance motion signal defined as y=0.01sin(1⋅2πt). To approximate the actual effect, random noise levels of 50 mV and 100 mV are added to the pressure sensor in order to examine the filter characteristics of the ESO. The simulation results illustrating the impact of noise interference are presented in the figure below.

As illustrated in Figure 21 and Figure 22, the introduction of random noise interference signals of 50 mV and 100 mV to the pressure sensor significantly affects the pressure readings. This impact is particularly pronounced at the peaks and troughs of the pressure curve. Figure 21 and Figure 22 illustrate that the maximum error fluctuations of the pressure curve prior to filtering are approximately 235 kPa and 655 kPa, respectively. As depicted in Figure 21, the pressure curve following filtering closely resembles the ideal pressure curve. Although Figure 22 indicates that the pressure curve after filtering exhibits some fluctuations, these variations remain within an acceptable range. The results indicate that when the random noise interference signal reaches 100 mV, the noise significantly affects the pressure; however, the filtering effect of the ESO remains effective. The filtered pressure curve is optimal, exhibiting smoothness and good tracking performance. After analyzing the noise filtering capabilities of the ESO, it can be concluded that the pressure values obtained after ESO filtering are suitable for use in pressure closed-loop control, thereby ensuring the system maintains robust closed-loop pressure control performance.

## 5. Conclusions

This paper establishes a state equation for the nonlinear mathematical model of the independent load port electro–hydraulic load simulation system. Based on this state equation, the controller is derived through a series of equations. To address the challenges of significant external motion disturbances and sensor noise interference experienced by the independent load port electro–hydraulic load simulation system, an integral sliding mode active disturbance rejection composite control strategy based on fixed-time convergence is proposed. The subsequent conclusions are drawn from the simulation research conducted:

(1)This paper proposes an independent load port electro–hydraulic load simulation system characterized by complex working conditions and multi-mode variable structures. The research findings indicate that this system offers a high degree of control freedom, rapid pressure response, and exceptional control accuracy. Its flexibility and convenience suggest that the independent load port electro–hydraulic load simulation system is well-suited for pressure loading control.(2)The simulation study demonstrates that when the disturbance signal amplitude is set to 10 mm and the frequency is 5 Hz, the independent load port electro–hydraulic load simulation system exhibits excellent anti-interference capability. This is achieved under the control of the integral sliding mode active disturbance rejection composite control strategy, which is based on fixed-time convergence. The results indicate that the closed-loop pressure immunity limit performance threshold of the system is relatively high, thereby confirming the effectiveness of the fixed-time convergence integral sliding mode auto-disturbance composite control strategy. Additionally, the independent load port electro–hydraulic load simulation system shows improved pressure control performance.(3)This paper explores the feasibility of nonlinear extended state observer technology for filtering. The simulation results demonstrate that the proposed integrated sliding mode active disturbance rejection composite control strategy, which is based on fixed-time convergence, exhibits effective suppression and filtering characteristics for random noise, thereby highlighting its superiority and practicality in pressure control applications.

This paper investigates the control mechanisms of the independent load port electro–hydraulic load simulation system, focusing specifically on the pressure control managed by a single servo valve. Under certain operating conditions, abrupt changes in the movement mode of the steering gear can lead to a mismatch between the electro–hydraulic load simulation system and the steering gear system. Such discrepancies may result in pressure shock issues. Therefore, there is an urgent need to enhance the mode-switching control strategy of the controller, warranting further in-depth research.

## Figures and Tables

**Figure 1 sensors-24-07400-f001:**
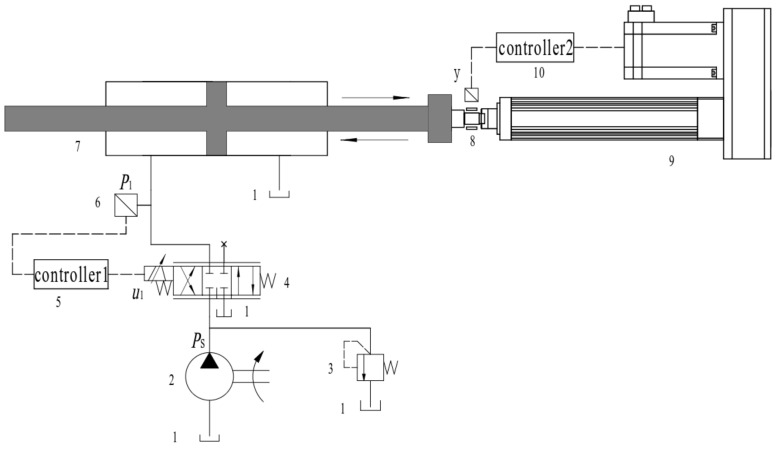
Schematic view of independent load port electro–hydraulic load simulation system model (1—tank, 2—hydraulic pump, 3—relief valve, 4—electro–hydraulic servo valve, 5—independent load port electro–hydraulic load simulation system controller, 6—pressure sensor, 7—hydraulic cylinder, 8—displacement sensor, 9—linear actuator, 10—linear actuator controller).

**Figure 2 sensors-24-07400-f002:**
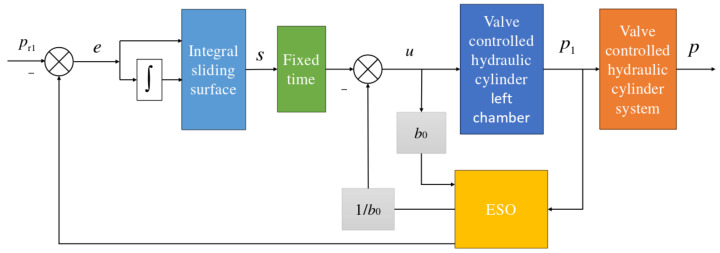
Block diagram of fixed-time integral sliding mode active disturbance rejection composite control.

**Figure 3 sensors-24-07400-f003:**
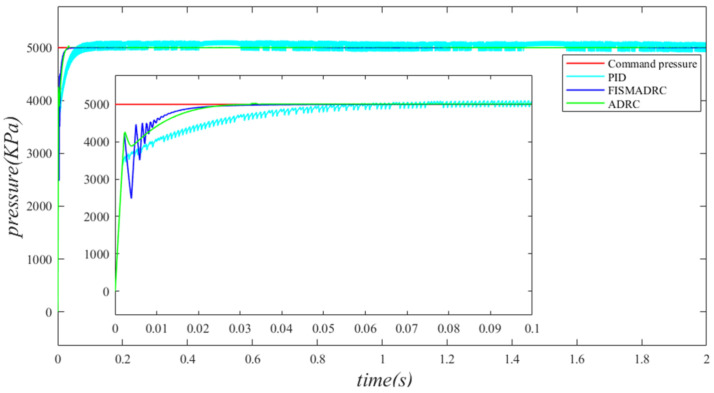
Closed-loop pressure with a disturbance frequency of 1 Hz.

**Figure 4 sensors-24-07400-f004:**
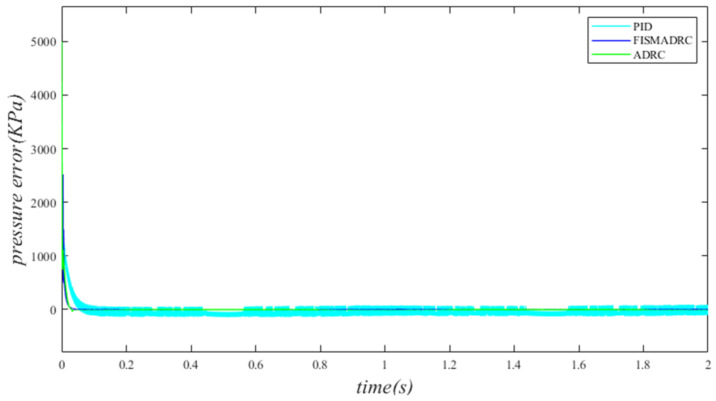
Closed-loop pressure error with a disturbance frequency of 1 Hz.

**Figure 5 sensors-24-07400-f005:**
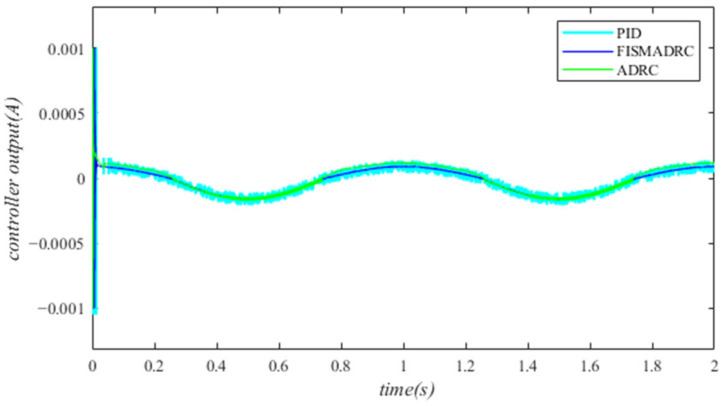
Controller output with a disturbance frequency of 1 Hz.

**Figure 6 sensors-24-07400-f006:**
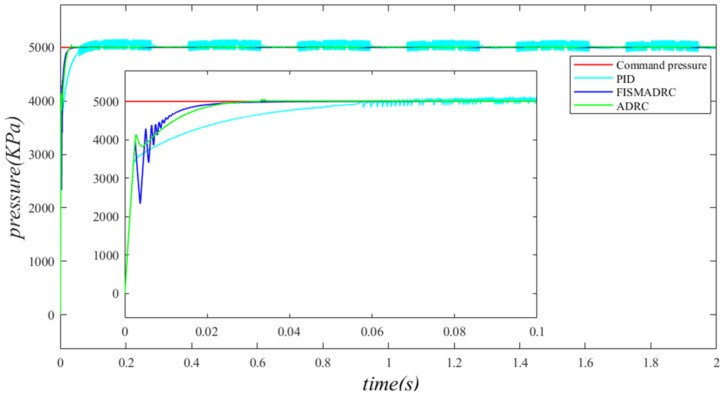
Closed-loop pressure with a disturbance frequency of 3 Hz.

**Figure 7 sensors-24-07400-f007:**
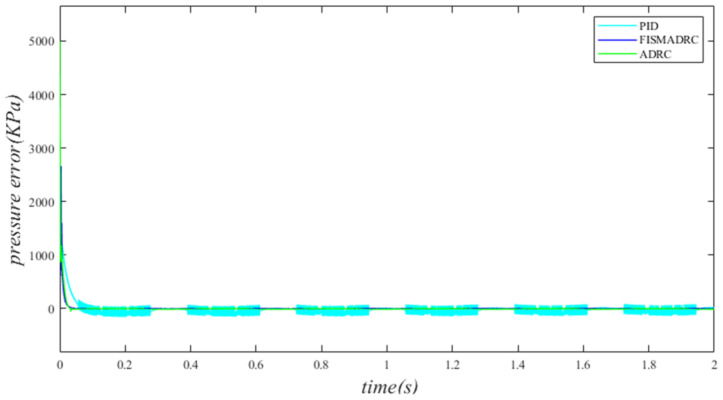
Closed-loop pressure error with a disturbance frequency of 3 Hz.

**Figure 8 sensors-24-07400-f008:**
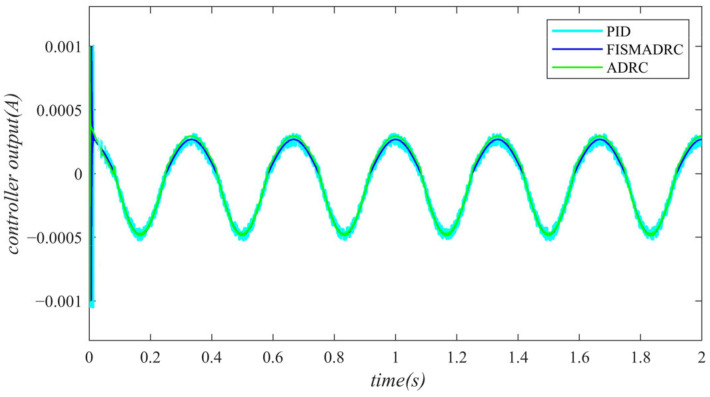
Controller output with a disturbance frequency of 3 Hz.

**Figure 9 sensors-24-07400-f009:**
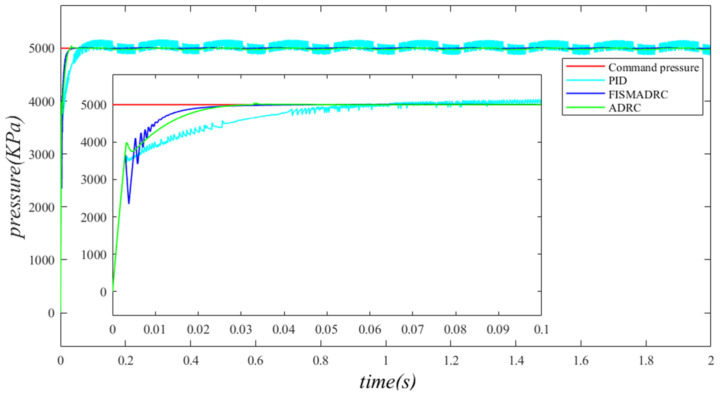
Closed-loop pressure with a disturbance frequency of 5 Hz.

**Figure 10 sensors-24-07400-f010:**
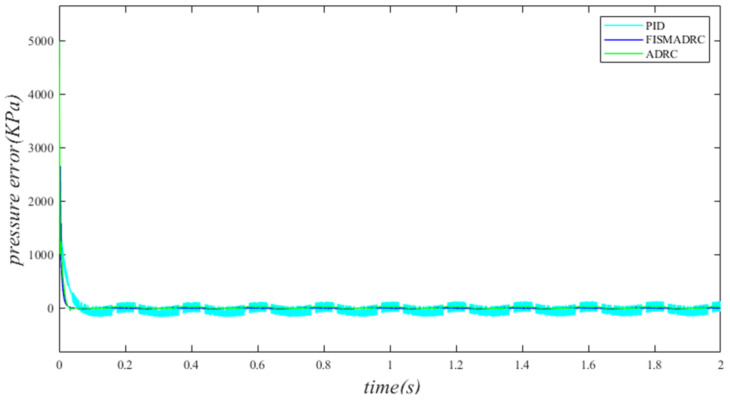
Closed-loop pressure error with a disturbance frequency of 5 Hz.

**Figure 11 sensors-24-07400-f011:**
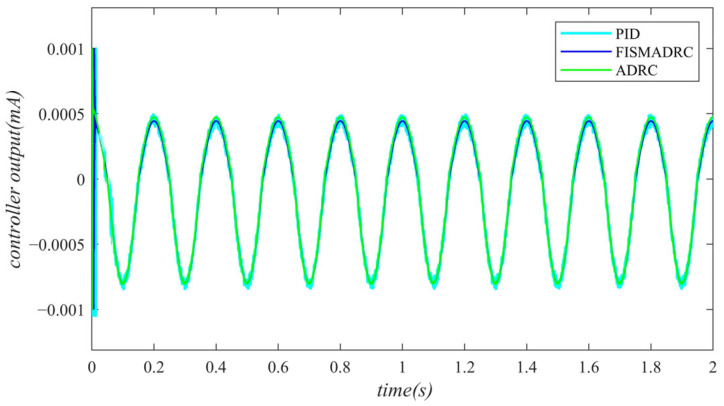
Controller output with a disturbance frequency of 5Hz.

**Figure 12 sensors-24-07400-f012:**
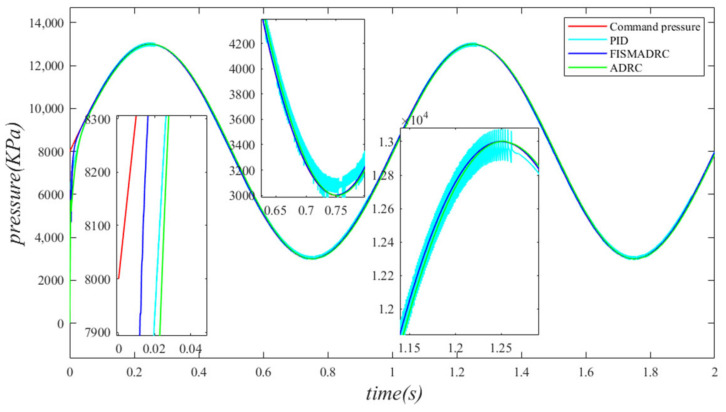
Closed-loop pressure with a disturbance frequency of 1 Hz.

**Figure 13 sensors-24-07400-f013:**
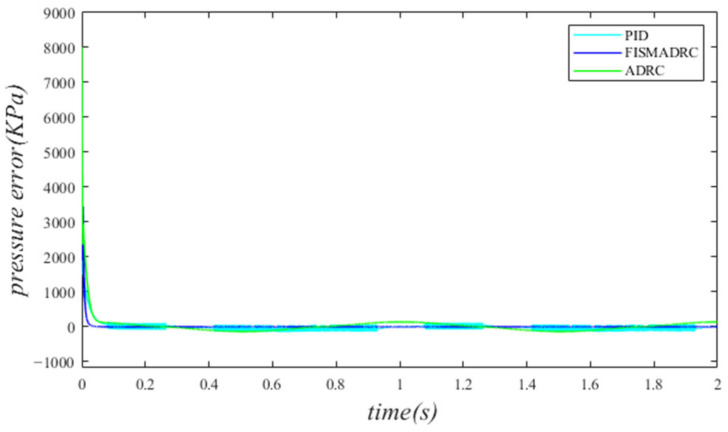
Closed-loop pressure error with a disturbance frequency of 1 Hz.

**Figure 14 sensors-24-07400-f014:**
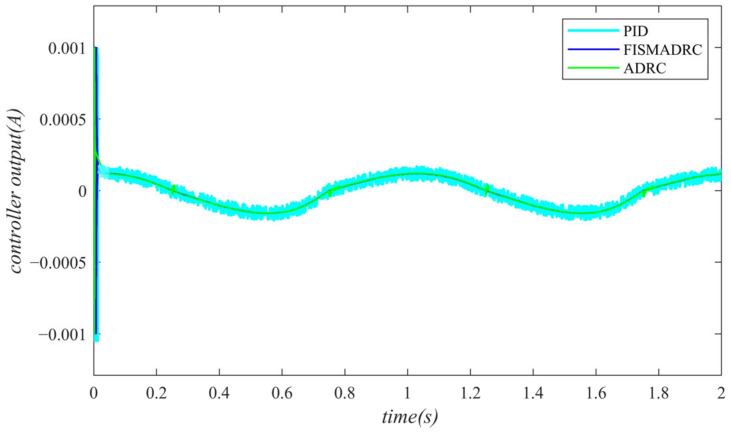
Controller output with a disturbance frequency of 1 Hz.

**Figure 15 sensors-24-07400-f015:**
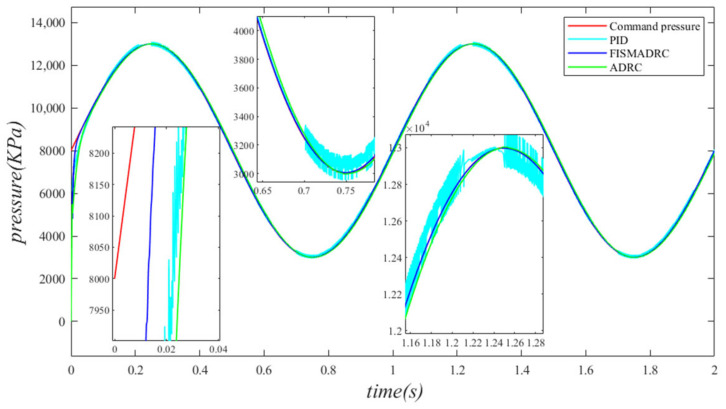
Closed-loop pressure with a disturbance frequency of 3 Hz.

**Figure 16 sensors-24-07400-f016:**
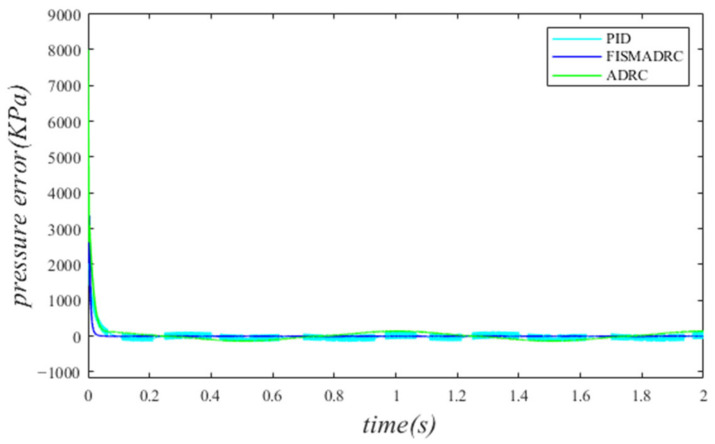
Closed-loop pressure error with disturbance frequency of 3 Hz.

**Figure 17 sensors-24-07400-f017:**
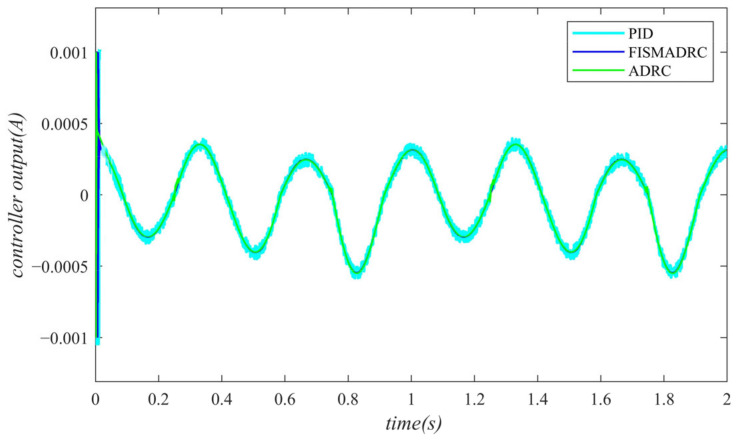
Controller output with a disturbance frequency of 3 Hz.

**Figure 18 sensors-24-07400-f018:**
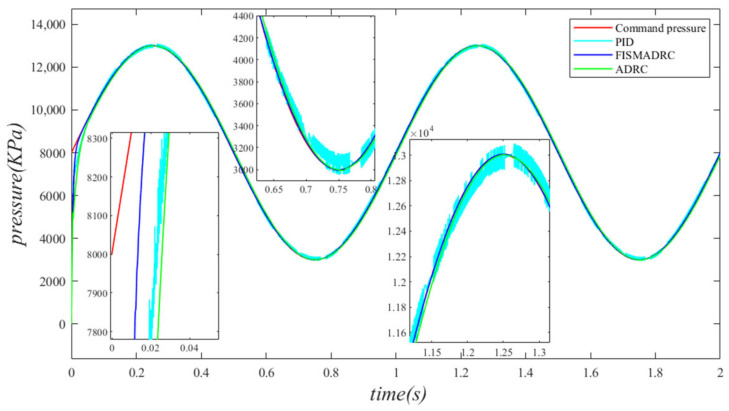
Closed-loop pressure with a disturbance frequency of 5 Hz.

**Figure 19 sensors-24-07400-f019:**
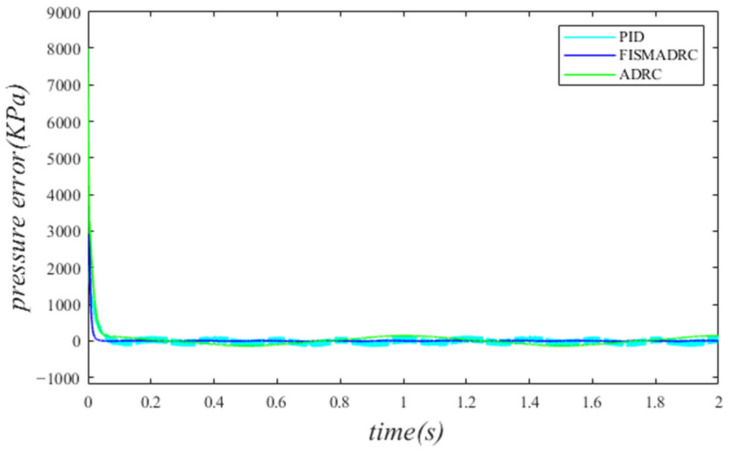
Closed-loop pressure error with a disturbance frequency of 5 Hz.

**Figure 20 sensors-24-07400-f020:**
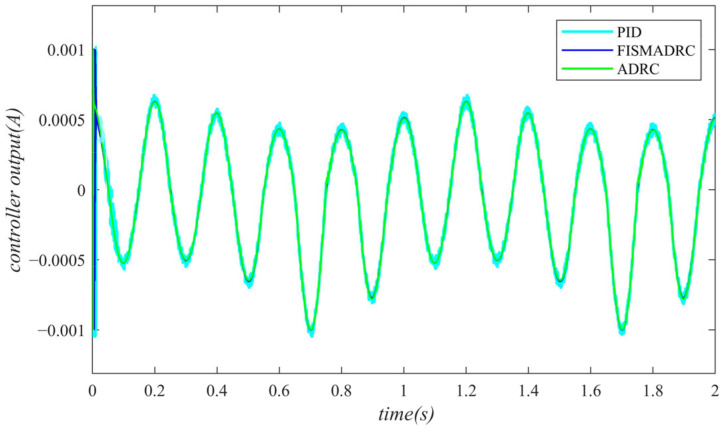
Controller output with a disturbance frequency of 5 Hz.

**Figure 21 sensors-24-07400-f021:**
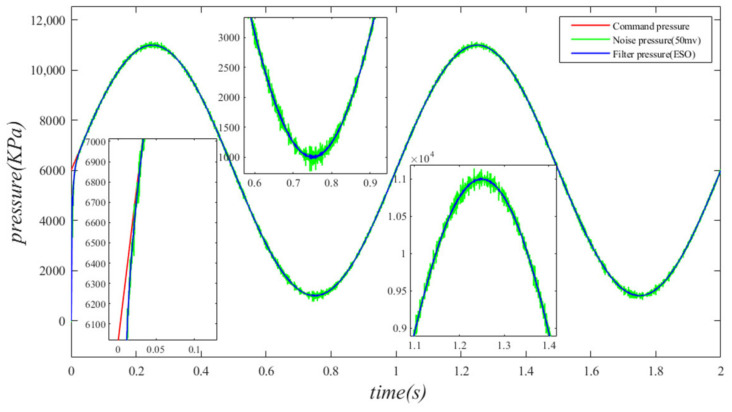
Pressure curve under random noise of 50 mV.

**Figure 22 sensors-24-07400-f022:**
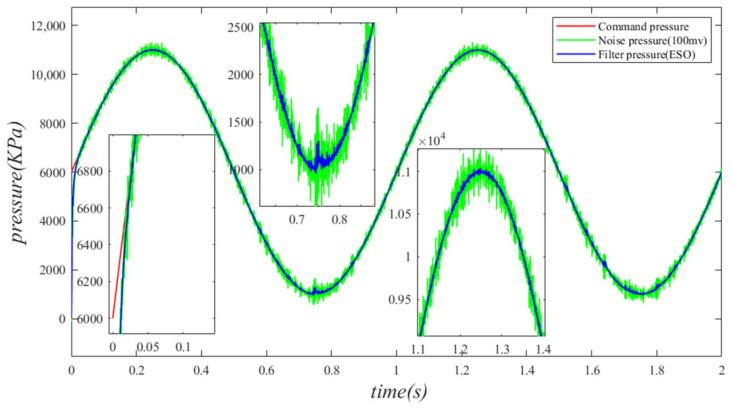
Pressure curve under random noise of 100 mV.

**Table 1 sensors-24-07400-t001:** System parameters.

Name	Numerical Value
Servo valve flow coefficient	$C_{d}=0.61$
Effective area of hydraulic cylinder piston rod	$A=0.003434m^{2}$
External leakage coefficient	$C_{tc}=2\times 10^{-15}{(m}^{3}/s)/\mathrm{Pa}$
Liquid volume elastic modulus of hydraulic fluids	$\beta_{e}=6.9\times 10^{8}\;\mathrm{Pa}$
Density of hydraulic fluid	$\rho =875{\;\mathrm{kg}/m}^{3}$
Hydraulic cylinder stroke	L=0.5 m
Maximum valve flow System pressure	117 L/min 21 MPa

**Table 2 sensors-24-07400-t002:** Controller simulation parameters.

FISMADRC				PID		ADRC	
$c_{1}$	0.1	$b_{0}$	$3.52\times 10^{7}\;$	P	15	$b_{0}$	$3.52\times 10^{7}\;$
$c_{2}$	1	$\beta_{1}$	$10^{3}\;$	I	5	$\beta_{1}$	$10^{3}\;$
m	1	$\beta_{2}$	$10^{6}\;$	D	0.1	$\beta_{2}$	$10^{6}\;$
n	0.5	δ	0.001			δ	0.001
ε	100	α	0.9			α	0.6
$g_{1}$	20	φ	1				
$g_{2}$	20	r	1.44				

## Data Availability

The authors confirm that the data supporting the findings of this study are available within the article.

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
