# Peer review of "Pressure Control of Multi-Mode Variable Structure Electro–Hydraulic Load Simulation System"

_sensors, 2024, doi:10.3390/s24227400_

Round 1

Reviewer 1 Report

Comments and Suggestions for Authors

1. Title should be rewritten and avoid "Research on"...

2. The abstract must provide a brief overview of the paper.  Thus, it should be re-written and should be to the point. What the problem is, what you did to solve it and what the results are and why are important?

3. Figure 1. Schematic view of independent......

4. Instead of "formula" use "equation". After the equations (3), (4), (8), (11), (13), (15), (16), (18) don't use "In the Formula" but "where".....

5. Please add a comparison to show how fast the pressure response is in relation to other systems.

6. Can you provide a metric to quantify the accuracy? Have you checked for percentage error or tolerance range data that could support the claim of 'exceptional' accuracy?

7. In conclusion section don't use "Article" but you can use "Paper" or "Research work"

Author Response

Thanks, "Please see the attachment."

Reviewer 2 Report

Comments and Suggestions for Authors

 This paper proposes an independent load port electro-hydraulic load simulation system, an composite control strategy about integral sliding mode active disturbance rejection   and  grounds  the active disturbance rejection control structure,  a series of innovative research works have been carried out,  good simulation test results are acquired. Papers research results would play an important role for to solve the significant positional disturbances present in the loading of electro-hydraulic loads,  effectively to mitigate  the impact uncertainty on the system peformance.

1. How to prove the stability and effectiveness of the integral sliding mode controller with fixed time convergence designed in the paper? 

2.In the simulation experiment, the selected interference frequencies are 1 Hz, 3 24 Hz, and 5 Hz. What is the reason for this?  if  use 50Hz,  What result will be?

3. The  design and strategy proposed in this  paper is only verified through simulation,  what problems may accure  in practical  system's applications?

Comments on the Quality of English Language

The English expression of the paper is good and appropriate,

Author Response

Thanks, "Please see the attachment."
